# ENSEMBLE METHODS AS A DEFENSE TO ADVERSARIAL PERTURBATIONS AGAINST DEEP NEURAL NETWORKS

## ABSTRACT

Deep learning has become the state of the art approach in many machine learning problems such as classification. It has recently been shown that deep learning is highly vulnerable to adversarial perturbations. Taking the camera systems of self-driving cars as an example, small adversarial perturbations can cause the system to make errors in important tasks, such as classifying traffic signs or detecting pedestrians. Hence, in order to use deep learning without safety concerns a proper defense strategy is required. We propose to use ensemble methods as a defense strategy against adversarial perturbations. We find that an attack leading one model to misclassify does not imply the same for other networks performing the same task. This makes ensemble methods an attractive defense strategy against adversarial attacks. We empirically show for the MNIST and the CIFAR-10 data sets that ensemble methods not only improve the accuracy of neural networks on test data but also increase their robustness against adversarial perturbations.

## 1 INTRODUCTION

In recent years, deep neural networks (DNNs) led to significant improvements in many areas ranging from computer vision (Krizhevsky et al., 2012; LeCun et al., 2015) to speech recognition (Hinton et al., 2012; Dahl et al., 2012). Some applications that can be solved with DNNs are sensitive from the security perspective, for example camera systems of self driving cars for detecting traffic signs or pedestrians (Papernot et al., 2016b; Sermanet & LeCun, 2011). Recently, it has been shown that DNNs can be highly vulnerable to adversaries (Szegedy et al., 2013; Goodfellow et al., 2014; Papernot et al., 2016a;b). The adversary produces some kind of noise on the input of the system to mislead its output behavior, producing undesirable outcomes or misclassification. Adversarial perturbations are carefully chosen in order to be hard, if not impossible, to be detected by the human eye (see figure 1). Attacks occur *after* the training of the DNN is completed. Furthermore, it has been shown that the exact structure of the DNN does not need to be known in order to mislead the system as one can send inputs to the unknown system in order to record its outputs to train a new DNN that imitates its behavior (Papernot et al., 2016b). Hence, in this manuscript it is assumed that the DNN and all its parameters are fully known to the adversary.

There are many methods on how to attack neural networks appearing in the literature. Some of the most well-known ones are the Fast Gradient Sign Method (Goodfellow et al., 2014) and its iterative extension (Kurakin et al., 2016), DeepFool (Moosavi-Dezfooli et al., 2016), Jacobian-Based Saliency Map Attack (Papernot et al., 2016c), and the L-BFGS Attack (Szegedy et al., 2013). This shows the need of building neural networks that are themselves robust against any kind of adversarial perturbations.

Novel methods on defending against adversarial attacks are appearing more and more frequently in the literature. Some of those defense methods are to train the network with different kinds of adversarially perturbated training data (Goodfellow et al., 2014; Papernot et al., 2016c), the use of distillation to reduce the effectiveness of the perturbation (Papernot et al., 2016d) or to apply denoising autoencoders to preprocess the data used by the DNN (Gu & Rigazio, 2014). It also has been noted that adversarial attacks can be detected (Metzen et al., 2017; Feinman et al., 2017), but

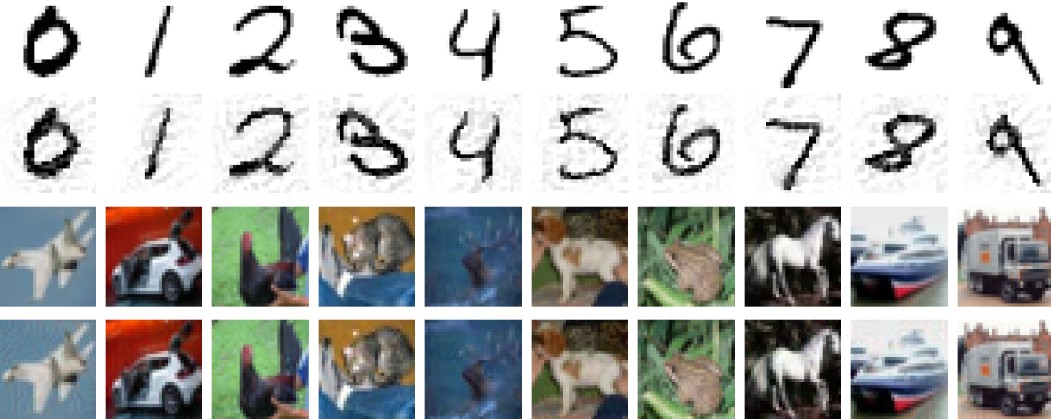

Figure 1: The first line shows original and correctly classified MNIST test data images. In the second line are the corresponding adversarial BIM attacks on a single classifier ($\epsilon = 0.2$, $\alpha = 0.025$, $n = 8$) which predicts (from left to right): 6, 8, 1, 5, 9, 3, 0, 2, 2, and 4. Analogously, the third line corresponds to correctly predicted examples of the CIFAR-10 test data set. In the bottom line are the corresponding adversarial BIM attacks on a single classifier ($\epsilon = 0.02$, $\alpha = 0.0025$, $n = 8$) which predicts (from left to right): deer, cat, deer, ship, bird, deer, deer, frog, automobile, and automobile.

these detection systems are again vulnerable to adversarial attacks. To our knowledge, there is no method that can reliably defend or detect *all* kinds of adversarial attacks.

In this manuscript, ensemble methods are used to obtain a classification system that is more robust against adversarial perturbations. The term ensemble method refers to constructing a set of classifiers used to classify new data points by the weighted or unweighted average of their predictions. Many ensemble methods have been introduced in the literature such as Bayesian averaging, Bagging (Breiman, 1996) and boosting (Dietterich et al., 2000). These methods frequently win machine learning competitions, for example the Netflix prize (Koren, 2009). Initial results on using ensembles of classifiers in adversarial context can be found in (Abbasi & Gagné, 2017; He et al., 2017). However, to the best of our knowledge this is the first manuscript that empirically evaluates the robustness of ensemble methods to adversarial perturbations.

One advantage of using ensemble methods as defense against adversarial perturbations is that they also increase the accuracy on unperturbed test data. This is not the case in general for other defense methods (see Table 4). However, in most applications a perturbated input can be considered as exception. Hence, it is desirable to obtain a state of the art result on unperturbed test data while making the model more robust against adversarial attacks. Another advantage is that ensemble methods can easily be combined with other defense mechanisms to improve the robustness against adversarial perturbations further (see Table 4). However, the advantages come at a cost of an increase of computational complexity and memory requirements which are proportional to the number of classifiers in the ensemble.

This paper is organized as follows: In section 2, some methods for producing adversarial perturbations are briefly introduced. Section 3 describes the defense strategy proposed in this manuscript. In section 4, the previous methods are tested on the MNIST and CIFAR-10 data sets and are compared to other defense strategies appearing in the literature. Finally, in section 5 the conclusions are presented.

## 2 ADVERSARIAL ATTACK

In this section, two methods for producing adversarial attacks shall be briefly described. In the following, let $\theta$ be the parameters of a model, $x$ the input of the model and $y$ the output value associated with the input value $x$. Further, let $J(\theta, x, y)$ be the cost function used to train the DNN.

FAST GRADIENT SIGN METHOD

The fast gradient sign method (FGSM) by Goodfellow et al. (2014) simply adds some small perturbations of size $\epsilon > 0$ to the input $x$,

$$x_{\text{FGSM}} = x + \epsilon \, \text{sign}[\nabla_x J(\theta, x, y)],$$

where the gradient $\nabla_x J(\theta, x, y)$ can be computed using backpropagation. This relatively cheap and simple adversarial perturbation performs well on many DNNs. It is believed that this behavior is due to linear elements such as ReLUs or maxout networks in the DNNs (Goodfellow et al., 2014).

BASIC ITERATIVE METHOD

The basic iterative method (BIM) by Kurakin et al. (2016) is an iterative extension of FGSM. The idea is to choose $\epsilon \geq \alpha > 0$ and then apply some perturbations similar to FGSM to the input $x$ and repeat the process $n$ times:

$$x_0 = x,$$
$$x_i = \text{clip}_{x,\epsilon}\left(x_{i-1} + \alpha \, \text{sign}[\nabla_{x_{i-1}} J(\theta, x_{i-1}, y)]\right),$$
$$x_{\text{BIM}} = x_n.$$

Here, $\text{clip}_{x,\epsilon}(\cdot)$ refers to clipping the values of the adversarial sample so that they remain within an $\epsilon$-neighborhood of $x$.

## 3 ENSEMBLE METHODS

Ensemble methods are widely used to improve classifiers in supervised learning (Dietterich et al., 2000). The idea is to construct a set of classifiers that is used to classify a new data point by the weighted or unweighted average of their predictions. In order for an ensemble to outperform a single classifier it must be both accurate and diverse (Hansen & Salamon, 1990). A classifier is said to be accurate if it is better than random guessing, and a set of classifiers is said to be diverse if different classifiers make different errors on new data points.

As expected, when performing adversarial perturbations on new data points different classifiers perform quite differently on these points. Hence, we conclude that diversity on adversarial perturbations is given. Furthermore, for adversarial perturbations with small $\epsilon > 0$, the vast majority of classifiers was accurate. In other words, for any small $\epsilon > 0$, we could not find an adversarial attack that would turn the majority of classifiers into non-accurate classifiers.

In section 4, the following ensemble methods are used. Note that random initialization of the model parameters is used in all methods.

(i) The first method is to train multiple classifiers with the same network architecture but with random initial weights. This results in quite diverse classifiers with different final weights (Kolen & Pollack, 1991).

(ii) The second method is to train multiple classifiers with different but similar network architectures to ensure obtaining a set of even more diverse classifiers. That is, extra filters are used in one classifier or an extra convolution layer is added to another classifier.

(iii) Third, *Bagging* (Breiman, 1996) is used on the training data. The term Bagging is derived from bootstrap aggregation and it consists of drawing $m$ samples with replacement from the training data set of $m$ data points. Each of these new data sets is called a bootstrap replicate. At average each of them contains $63.2\%$ of the training data, where many data points are repeated in the bootstrap replicates. A different bootstrap replicate is used as training data for each classifier in the ensemble.

(iv) The last method is to add some small Gaussian noise to the training data so that all classifiers are trained on similar but different training sets. Note that adding Gaussian noise to the training data also makes each classifier somewhat more robust against adversarial perturbations.

Once an ensemble of classifiers is trained, it predicts by letting each classifier vote for a label. More specifically, the predicted value is chosen to be the label that maximizes the average of the output probabilities from the classifiers in the ensemble.

In order to attack a network with the methods from section 2 the gradient $\nabla_x J(\theta, x, y)$ must be computed. However, obtaining the gradient for an ensemble requires to calculate the gradient of each of its classifiers. Nevertheless, the following two methods are used to estimate the gradient of an ensemble, which are referred to as Grad. 1 and Grad. 2 for the rest of this manuscript:

**Grad. 1** Use $\nabla_x J(\theta_i, x, y)$ of the $i$-th classifier. This is clearly not the correct gradient for an ensemble. But the question is whether an attack with this gradient can already mislead all classifiers in the ensemble in a similar manner.

**Grad. 2** Compute the average of the gradients $\frac{1}{n} \sum_i \nabla_x J(\theta_i, x, y)$ from all classifiers in the ensemble.

A comparison of the effects of these two gradients for attacking ensembles can be found in section 4.

## 4 EXPERIMENTS

In this section the ensemble methods from section 3 are empirically evaluated on the MNIST (LeCun et al., 1998) and the CIFAR-10 (Krizhevsky & Hinton, 2009) data sets which are scaled to the unit interval. All experiments have been performed on ensembles of 10 classifiers. Note that this choice has been done for comparability. That is, in some cases the best performance was already reached with ensembles of less classifiers while in others more classifiers might improve the results.

A summary of the experimental results can be found in Table 2 and the corresponding visualization in Figure 2. A comparison of ensembles with other defense methods and a combination of those with ensembles can be found in Table 4. In the following all FGSM perturbations are done with $\epsilon = 0.3$ on MNIST and with $\epsilon = 0.03$ on CIFAR-10. Furthermore, all BIM perturbations are done with $\epsilon = 0.2$, $\alpha = 0.025$ and $n = 8$ iterations on MNIST and with $\epsilon = 0.02$, $\alpha = 0.0025$ and $n = 8$ on CIFAR-10. The abbreviations in Table 2 and in Figure 2 shall be interpreted in the following way: Rand. Ini. refers to random initialization of the weights of the neural network, Mix. Mod. means that the network architecture was slightly different for each classifier in an ensemble, Bagging refers to classifiers trained on bootstrap replicates of the training data, and Gauss noise implies that small Gaussian noise has been added to the training data. Each ensemble is attacked with FGSM and BIM based on the gradients from Grad. 1 and Grad. 2. In Table 2, the term Single refers to evaluating a single classifier.

### MNIST

The MNIST data set consists of 60,000 training and 10,000 test data samples of black and white encoded handwritten digits. The objective is to classify these digits in the range from 0 to 9. A selection of images from the data set and some adversarial perturbations can be found in the top two rows of figure 1. In the experiments, the network architecture in Table 1 is used and it is trained with 10 epochs. All results from the experiments are summarized in Table 2.

On unperturbed test data the classification accuracy is roughly 99%. The difference between single classifiers and ensembles is below one percent throughout. The ensembles slightly outperform the single classifiers in all cases.

Table 1: MNIST Network Architecture

| Layer Type | Parameters |
| --- | --- |
| Relu Convolutional | 32 filters ($3 \times 3$) |
| Relu Convolutional | 32 filters ($3 \times 3$) |
| Max Pooling | $2 \times 2$ |
| Relu Fully Connected | 128 units |
| Dropout | 0.5 |
| Relu Fully Connected | 10 units |
| Softmax | 10 units |

Table 2: Experimental results on the MNIST and the CIFAR-10 data sets

MNIST Accuracy

| Test Data | | No Attack | | Grad. 1 | | Grad. 2 |
|---|---|---|---|---|---|---|
| Type | Method | Single | Ensemble | Single | Ensemble | Ensemble |
| FGSM | Rand. Ini. | 0.9912 | 0.9942 | 0.3791 | 0.6100 | 0.4517 |
| | Mix. Mod. | 0.9918 | 0.9942 | 0.3522 | 0.5681 | 0.4609 |
| | Bagging | 0.9900 | 0.9927 | 0.4045 | 0.6738 | 0.5716 |
| | Gauss Noise | 0.9898 | 0.9920 | 0.5587 | 0.7816 | 0.7043 |
| BIM | Rand. Ini. | 0.9912 | 0.9942 | 0.0906 | 0.6518 | 0.8875 |
| | Mix. Mod. | 0.9918 | 0.9942 | 0.0582 | 0.6656 | 0.9076 |
| | Bagging | 0.9900 | 0.9927 | 0.1110 | 0.7068 | 0.9233 |
| | Gauss Noise | 0.9898 | 0.9920 | 0.5429 | 0.9152 | 0.9768 |

CIFAR-10 Accuracy

| Test Data | | No Attack | | Grad. 1 | | Grad. 2 |
|---|---|---|---|---|---|---|
| Type | Method | Single | Ensemble | Single | Ensemble | Ensemble |
| FGSM | Rand. Ini. | 0.7984 | 0.8448 | 0.1778 | 0.4538 | 0.3302 |
| | Mix. Mod. | 0.7898 | 0.8400 | 0.1643 | 0.4339 | 0.3140 |
| | Bagging | 0.7815 | 0.8415 | 0.1822 | 0.4788 | 0.3571 |
| | Gauss Noise | 0.7160 | 0.7687 | 0.2966 | 0.6097 | 0.4707 |
| BIM | Rand. Ini. | 0.7984 | 0.8448 | 0.1192 | 0.5232 | 0.6826 |
| | Mix. Mod. | 0.7898 | 0.8400 | 0.1139 | 0.5259 | 0.6768 |
| | Bagging | 0.7815 | 0.8415 | 0.1280 | 0.5615 | 0.7166 |
| | Gauss Noise | 0.7160 | 0.7687 | 0.3076 | 0.6735 | 0.7277 |

This picture changes dramatically if the networks are attacked by one of the methods described in section 2. Using the FGSM attack with gradients from Grad. 1 on a single classifier, the classification rate drops down to a range of roughly 35%–56%. The ensembles perform significantly better by producing an accuracy of 57%–78%. Evaluating the same with gradients from Grad. 2 it turns out that ensemble methods still obtain an accuracy of 45%–70%. The higher accuracy of Grad. 1 is expected since in contrast to Grad. 2 it computes the gradients with respect to just one classifier. Nevertheless, the ensembles outperform single classifiers in each case by approximately 7%-22%.

The decrease of the accuracy is even more extreme for single classifiers if the BIM method is used. Here, the accuracy can be as low as around 6% and only the classifiers trained with Gaussian noise significantly exceed the 10%. The accuracy of the ensemble methods against attacks using Grad. 1 is considerably higher with 65%–92%. Furthermore, ensembles are even more robust against BIM attacks based on Grad. 2 with a correct classification rate of 89%–98%. It is surprising that BIM attacks using Grad. 1 are more successful than those using Grad. 2, because Grad. 1 only attacks a single classifier in the ensemble. Concluding, the ensemble methods outperform single classifiers significantly by 37%-85% on BIM attacks.

Focusing on the different defense strategies, we observe that using random initialization of the network weights as well as using several networks of similar architectures for an ensemble generally improves the robustness against adversarial attacks considerably in comparison with single classifiers. Bagging outperforms both of the previous methods on adversarial perturbations, but performs slightly worse on unperturbed test data. Using ensembles with small Gaussian noise on the training data results in the best defense mechanism against adversarial attacks. This may be due to the fact that using additive noise on the training data already makes every single classifier in the ensemble more robust against adversarial perturbations. On the down-side, adding Gaussian noise to the

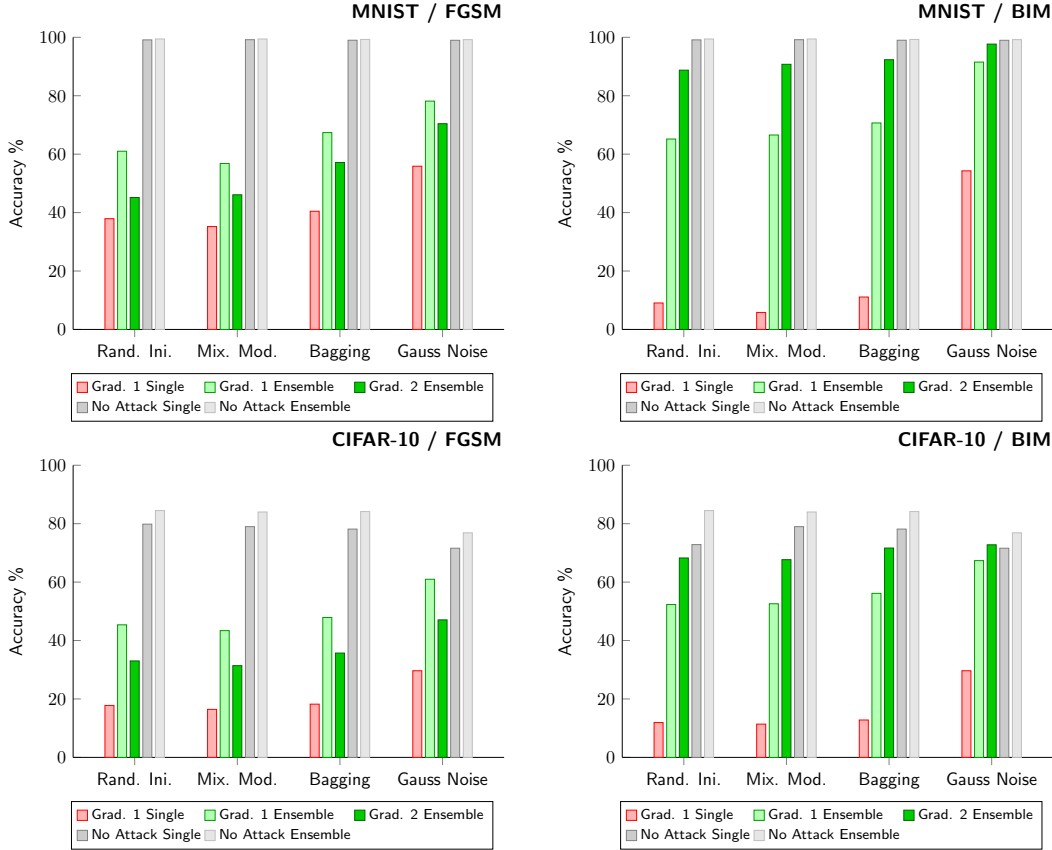

Figure 2: Visual comparisons of the accuracies presented in Table 2. Compared are the MNIST (top row) and CIFAR-10 (bottom row) data sets on the FGSM (left column) and the BIM (right column) attacks. Grad. 1 Single refers to attacks based on Grad. 1 on single classifiers, Grad. 1 Ensemble refers to attacks based on Grad. 1 on ensembles, Grad. 2 Ensemble refers to attacks based on Grad. 2 on ensemble classifiers, No Attack Single refers to single classifier on unperturbed data, and finally No Attack Ensemble refers to ensemble classifiers on unperturbed data.

training data performs worst from all considered ensemble methods on test data. However, such an ensemble still performs better than all single classifiers on MNIST.

CIFAR-10

The CIFAR-10 data set consists of 50,000 training and 10,000 test data samples of three-color component encoded images of ten mutually exclusive classes: airplane, automobile, bird, cat, deer, dog, frog, horse, ship, and truck. A selection of images from the data set and some adversarial perturbations can be found in the two bottom rows of figure 1. In all experiments the network architecture described in Table 3 is used and the networks are trained with 25 epochs.

In general, the observations on the MNIST data set are confirmed by the experiments on CIFAR-10. Since the latter data set is more demanding to classify, the overall classification rate is already lower in the attack-free case, where single classifiers reach an accuracy of roughly 72%–80%, while ensembles show a higher accuracy of 77%–84%. Note that there are network architectures in the literature that outperform our classifiers considerably on test data (Graham, 2014).

The FGSM attacks on single classifiers using method Grad. 1 show a drop-down of the accuracy to 16%-30%. In contrast, ensembles are significantly better reaching accuracies of 43%-61% when attacked using Grad. 1 and 31%-47% when attacked with Grad. 2.

Table 3: CIFAR-10 Network Architecture

| Layer Type | Parameters |
|---|---|
| Relu Convolutional | 32 filters (3×3) |
| Relu Convolutional | 32 filters (3×3) |
| Max Pooling | 2×2 |
| Dropout | 0.2 |
| Relu Convolutional | 64 filters (3×3) |
| Relu Convolutional | 64 filters (3×3) |
| Max Pooling | 2×2 |
| Dropout | 0.3 |
| Relu Convolutional | 128 filters (3×3) |
| Relu Convolutional | 128 filters (3×3) |
| Max Pooling | 2×2 |
| Dropout | 0.4 |
| Relu Fully Connected | 512 units |
| Dropout | 0.5 |
| Relu Fully Connected | 10 units |
| Softmax | 10 units |

When using BIM attacks accuracies for single classifiers lie between 11% and 31%. Again, the ensemble methods outperform the single classifiers reaching accuracies of 52%-67% when attacked using Grad. 1 and 68%-73% when attacked with Grad. 2.

The same observations as on the MNIST data set can be made on the CIFAR-10 data set. All ensemble methods outperform single classifiers when comparing their robustness against adversarial perturbations. FGSM attacks on an ensemble using Grad. 2 outperform those using Grad. 1, as expected. Similar to the MNIST experiments, when using BIM attacks, ensembles are surprisingly more robust against gradient attacks from Grad. 2 than against gradient attacks from Grad. 1. The reason for this might be that the gradient portion from different classifiers using Grad. 2 in the ensemble try to reach a different local maximum and block each other in the following iterations.

As already observed on the MNIST data set, Bagging performs better than random initialization and than using similar but different network architectures. Again, adding small Gaussian noise on the training data performs best on adversarial perturbations but relatively poor on real test data on CIFAR-10.

COMPARISON WITH OTHER METHODS

In this section, we compare the previous results with two of the most popular defense methods: adversarial training (Goodfellow et al., 2014; Papernot et al., 2016c) and defensive distillation (Papernot et al., 2016d). Furthermore, we show the positive effects of combining those methods with ensembles. For simplicity, we only consider the gradient Grad. 2 whenever an ensemble is attacked. The results are summarized in Table 4. Here, the content shall be interpreted in the following way: Bagging refers to ensembles trained with bagging, Adv. Train. to adversarial training, Def. Dist. to defensive distillation, the operator $+$ to combinations of the previous methods, bold text to the best performance of the first three methods, and the asterisk to the best method including combinations of defensive strategies.

Adversarial training (AT) is a method that uses FGSM as regularizer of the original cost function:

$$J_{AT}(\theta, x, y) = \rho J(\theta, x, y) + (1 - \rho)J(\theta, x + \epsilon \operatorname{sign}(\nabla_x J(\theta, x, y)), y),$$

where $\rho \in [0, 1]$. This method iteratively increases the robustness against adversarial perturbations. In our experiments, we use $\rho = \frac{1}{2}$ as proposed in Goodfellow et al. (2014).

Table 4: Accuracies of different defense mechanisms

| Methods | MNIST | | | CIFAR-10 | | |
|---|---|---|---|---|---|---|
| | No Attack | FGSM | BIM | No Attack | FGSM | BIM |
| Bagging | **0.9927**$^*$ | **0.5716** | **0.9233** | **0.8415**$^*$ | **0.3571** | **0.7166**$^*$ |
| Adv. Train. | 0.9902 | 0.3586 | 0.5420 | 0.7712 | 0.1778 | 0.3107 |
| Def. Dist. | 0.9840 | 0.0798 | 0.3829 | 0.7140 | 0.1828 | 0.3635 |
| Bagging + Adv. Train. | 0.9927$^*$ | 0.8703* | 0.9840* | 0.8320 | 0.5010* | 0.7017 |
| Bagging + Def. Dist. | 0.9875 | 0.0954 | 0.4514 | 0.7323 | 0.1839 | 0.4569 |

In defensive distillation a teacher model $F$ is trained on a training data set $X$. Then smoothed labels at temperature $T$ are computed by

$$F^T(X) = \left[ \frac{\exp(F_i(X)/T)}{\sum_{i=1}^{N} \exp(F_i(X)/T)} \right]_{i \in \{1,\dots,N\}},$$

where $F_i(X)$ refers to the probability of the $i$-th out of $N$ possible classes. A distilled network is a network that is trained on the training data $X$ using the smoothed labels $F^T(X)$. In the following, we use $T = 10$ based on the experimental results in Papernot et al. (2016d).

We found that single networks trained with adversarial training or defensive distillation have a lower accuracy than ensembles trained with bagging (see the top three rows in Table 4). This is not only the case on the considered attacked data but also on unperturbated test data. Combining ensembles with adversarial training can improve the robustness against adversarial perturbations further, while a combination with defensive distillation does not reveal the same tendency (see the two bottom rows in Table 4). We emphasize that already the standard ensemble method does not only outperform both adversarial training and defensive distillation throughout but also has the overall highest accuracy on unperturbated test data.

## 5 CONCLUSION

With the rise of deep learning as the state-of-the-art approach for many classification tasks, researchers noted that neural networks are highly vulnerable to adversarial perturbations. This is particularly problematic when neural networks are used in security sensitive applications such as autonomous driving. Hence, with the development of more efficient attack methods against neural networks it is desirable to obtain neural networks that are themselves robust against adversarial attacks.

In this manuscript, it is shown that several ensemble methods such as random initialization or Bagging do not only increase the accuracy on the test data, but also make the classifiers considerably more robust against certain adversarial attacks. We consider ensemble methods as sole defense methods, but more robust classifiers can be obtained by combining ensemble methods with other defense mechanisms such as adversarial training. Although only having tested simple attack scenarios, it can be expected that ensemble methods may improve the robustness against other adversarial attacks.

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
