# OpenReview forum: "Ensemble Methods as a Defense to Adversarial Perturbations Against Deep Neural Networks"
_ICLR.cc/2018/Conference — Reject_

### Official Review · AnonReviewer2 · 2017-11-24
**A simple technique missing comparison with related ones.**

**Rating:** 7
**Confidence:** 3

**Review:**

This paper describes the use of ensemble methods to improve the robustness of neural networks to adversarial examples. Adversarial examples are images that have been slightly modified (e.g. by adding some small perturbation) so that the neural network will predict a wrong class label.

Ensemble methods have been used by the machine learning community since long time ago to provide more robust and accurate predictions.

In this paper the authors explore their use to increase the robustness of neural networks to adversarial examples.

Different ensembles of 10 neural networks are considered. These include techniques such as bagging or injecting noise in the
training data.

The results obtained show that ensemble methods can sometimes significantly improve the robustness against adversarial examples. However,
the performance of the ensemble is also highly deteriorated by these examples, although not as much as the one of a single neural network.

The paper is clearly written.

I think that this is an interesting paper for the deep learning community showing the benefits of ensemble methods against adversarial
examples. My main concern with this paper is the lack of comparison with alternate techniques to increase the robustness against adversarial examples. The authors should have compared with the methods described in:

(Goodfellow et al., 2014; Papernot et al., 2016c),
(Papernot et al., 2016d)
(Gu & Rigazio, 2014)

Furthermore, the ensemble approach has the main disadvantage of increasing the prediction time by a lot. For example, with 10 elements in the ensemble, predictions are 10 times more expensive.
------------------------------
I have read the updated version of the paper. I think the authors have done a good job comparing with related techniques. Therefore, I have slightly increased my score.

---

> ### Author Response · Authors · 2017-12-18
> **We highly appreciate your feedback**
>
> We highly appreciate your feedback.
>
> In respect to your concerns about the comparability with other methods: We added a section where we compare ensembles with “adversarial learning” (Goodfellow et al., 2014; Papernot et al., 2016c) and with “defensive distillation” (Papernot et al., 2016d). We hope that this resolves your concerns about comparability. Note, we did not compare with (Gu & Rigazio, 2014), due to the non-trivial parameter choices required by this method (particularly the choice of the network architecture).
>
> We agree with you that the increased computational time when using ensembles should be mentioned in the paper. Hence, we added a new paragraph to the manuscript about the advantages and disadvantages of using ensembles including topics like prediction time, memory requirements, but also higher accuracy on unperturbed test data (we found this is one of the main advantages of ensembles over other defense methods).

---

### Official Review · AnonReviewer3 · 2017-11-26
**A solid and effective idea, but a limited analysis.**

**Rating:** 5
**Confidence:** 3

**Review:**

Summary: This paper proposes to use ensembling as an adversarial defense mechanism. The defense is evaluated on MNIST and CIFAR10 ans shows reasonable performance against FGSM and BIM.

Clarity: The paper is clearly written and easy to follow.

Originality: Building an ensemble of models is a well-studied strategy that was shown long ago to improve generalization. As far as I know, this paper is however the first to empirically study the robustness of ensembles against adversarial examples.

Quality: While this paper contributes to show that ensembling works reasonably well against adversarial examples, I find the contribution limited in general.
- The method is not compared against other adversarial defenses.
- The results illustrate that adding Gaussian noise on the training data clearly outperforms the other considered ensembling strategies. However, the authors do not go beyond this observation and do not appear to try to understand why it is the case.
- Similarly, the Bagging strategy is shown to perform reasonably well (although it appears as a weaker strategy than Gaussian noise) but no further analysis is carried out. For instance, it is known that the reduction of variance is maximal in an ensemble when its constituents are maximally decorrelated. It would be worth studying more systematically if this correlation (or 'diversity') has an effect on the robustness against adversarial examples.
- I didn't understand the motivation behind considering two distinct gradient estimators. Why deriving the exact gradient of an ensemble is more complicated?

Pros:
- Simple and effective strategy.
- Clearly written paper.
Cons:
- Not compared against other defenses.
- Limited analysis of the results.
- Ensembling neural networks is very costly in terms of training. This should be considered.

Overall, this paper presents an interesting and promising direction of research. However, I find the current analysis (empirically and/or theoretically) to be too limited to constitutes a solid enough piece of work. For this reason, I do not recommend this paper for acceptance.

---

> ### Author Response · Authors · 2017-12-18
> **We greatly appreciate your insightful feedback**
>
> We greatly appreciate your insightful feedback. We would like to respond to your comments concerning quality:
>
> 1.	We added a comparison with other defense methods, specifically with “adversarial training” (Goodfellow et al., 2014; Papernot et al., 2016c) and with “defensive distillation” (Papernot et al., 2016d).
>
> 2.	It is true that adding Gaussian noise produced the best defense strategy, however this came at a cost of a reduction in accuracy on (unperturbed) test data of about 7% in the Cifar-10 case. That is why we considered Bagging as the better method: it might be a little worse on adversarial perturbed data but better than the Gaussian noise case on unperturbed test data. It is our believe that in real applications unperturbed data is the standard case and adversarial attacked data is a special event. Hence, loosing accuracy on test data can be quite problematic.
>
> 3.	Thanks for your idea of evaluating the effect of diversity of the classifiers on the defensive performance of the ensembles. We think that this is worth looking into. But we believe, this would go beyond the scope of our manuscript.
>
> 4.	The objective of using Grad. 1 was to study the transferability of an attack of one classifier to all classifiers in the ensemble. As we mentioned in the paper, Grad. 2 represents the correct gradient to attack an ensemble.
>
> 5.	We agree with you that computing Grad. 1  is no more complicated than computing Grad. 2. Hence, we changed the corresponding sentences accordingly.
>
> You are correct about the increased computational costs when using ensembles. We therefore added a new paragraph were we highlight the advantages of using ensembles as well as the disadvantages (like an increase of computational costs and memory requirements). The advantage section includes especially the increase in accuracy on unperturbated test data while still performing well against adversaries.

---

### Official Review · AnonReviewer1 · 2017-12-03
**Simple but effective idea, light in the presentation and experimentations**

**Rating:** 4
**Confidence:** 4

**Review:**

In this manuscript, the authors empirically investigated the robustness of some different deep neural networks ensembles to two types of attacks, namely FGSM and BIM, on two popular datasets, MNIST and CIFAR10. The authors concluded that the ensembles are more accurate on both clean and adversaries samples than a single deep neural network. Therefore, the ensembles are more robust in terms of the ability to correctly classify the adversary attacks.

As the authors stated, an attack that is designed to fool one network does not necessarily fool the other networks in the same way. This is likely why ensembles appear more robust than single deep learners. However, robustness of ensembles to the white-box attacks that are generated from the ensemble is still low for FGS. Generally speaking, although FGS attacks generated from one network can fool less the whole ensembles, generating FGS adversaries from a given ensemble is still able to effectively fool it. Therefore, if the attacker has access to the ensemble or even know the classification system based on that ensemble, then the ensemble-based system is still vulnerable to the attacks generated specifically from it. Simple ensemble methods are not likely to confer significant robustness gains against adversaries.

In contrast to FGS results, surprisingly BIM-Grad1 is able to fool more the ensemble than BIM-Grad2. Therefore, it seems that if the attacker makes BIM adversaries from only a single classifier, then she can simply and yet effectively mislead the whole ensemble. In comparison to BIM-Grad2, BIM-Grad1 results show that BIM attacks from one network (BIM-Grad1) can more successfully fool the other different networks in the ensembles in a similar way! BIM-Grad2 is not that much able to fool the ensemble-based system even this attack generated from the ensemble (white-box attacks). In order to confirm the robustness of the ensembles to BIM attacks, the authors can do more experiments by generating BIM-Grad2 attacks with higher number of iterations.

Indeed, the low number of iterations might cause the lower rate of success for generating adversaries by BIM-Grad2. In fact, BIM adversaries from the ensembles might require more number of iterations to effectively fool the majority of the members in the ensembles. Therefore, increasing the number of iterations can increase the successful rate of generating BIM-Average Grad2 adversaries. Note that in this case, it is recommended to compare the amount of distortion (perturbation) with different number of iterations in order to indicate the effectiveness of the ensembles to white-box BIM attacks.

Despite to averaging the output probabilities to compute the ensemble final prediction, the authors generated the adversaries from the ensemble by computing the sum of the gradients of the classifiers loss. A proper approach would have been to average of these gradients. The fact the sum is not divided by the number of members (i.e., sum of gradients instead of average of gradients) is increasing the step size of the adversarial method proportionally to the ensemble size, raising questions on the validity of the comparison with the single-model adversarial generation.

Overall, I found the paper as having several methodological flaws in the experimental part, and rather light in terms of novel ideas. As noticed in the introduction, the idea of using ensemble for enhancing robustness as already been proposed. Making a paper only to restate it, is too light for acceptation. Moreover, experimental setup using a lot of space for comparing results on standard datasets (i.e., MNIST and CIFAR10), even with long presentation of these datasets. Several issues are raised in the current experiments and require adjustments. Experiments should also be more elaborated to make the case stronger, following at least some of indications provided.

---

> ### Author Response · Authors · 2017-12-18
> **Thank you very much for your constructive feedback**
>
> Thank you very much for your constructive feedback and your valuable comments.
>
> We first like to respond to your last comment about originality of our work: To the best of our knowledge, this is the first paper to empirically evaluate the robustness of ensembles against adversarial attacks. The first paper we cited in this context was about how to build ensembles of specialist defenses (classifiers that classify on a subset of classes only and then are joined to be able to predict all classes. The focus is rather on how to build these specialist classifiers.) and the second paper showed an attack on how to break such specialist defenses. However, we did not find any paper that considered general ensemble methods as defense mechanism and analyzed what kind of ensembles are more robust.
>
> We agree with you in terms that by attacking with FGSM in combination with Adv. 2 and BIM in combination with Adv. 1 one obtains the strongest attack against ensembles, which we also wrote in the experimental part of our paper.
>
> To your comment that the accuracy on attacked ensembles is relatively low we would like to highlight that all images were scaled to the unit interval [0,1]. Hence, for example the BIM attack on MNIST could make a maximum distortion of 20% at each pixel and in CIFAR-10 case up to 2%. We added a comparison of our method with other defense methods (defensive distillation and adversarial training) on the same kind of attacks to show the effectiveness of ensembles.
>
> In respect to running BIM-Grad. 2 attacks with more iterations: You are correct that by increasing the number of iterations one can get somewhat better attacks (however this comes at a significant increase of computational cost). Nevertheless, we had to fix the parameters for our evaluations. Note that in the BIM attack the values are clipped to be in an epsilon neighborhood of the true image. This might be why running the attacks for more iterations has no major effect.
>
> You mentioned that in Grad. 2 the average of the gradients might be better than the sum of the gradients. Here, we like to point out that in both the FGSM and the BIM attack one always computes the sign function of the gradients and sign(\sum(gradients)) = sign(\average(gradients)). However, we agree with you that the average is the correct gradient for our weighting system (even though it results in the very same FGSM and BIM attacks). Hence, we changed our manuscript accordingly.
>
> To your final comment about the experiment part: we added a comparison of our method with “defensive distillation” (Papernot et al., 2016d) and “adversarial training” (Goodfellow et al., 2014; Papernot et al., 2016c) to make our case stronger.

---

### Author Response · Authors · 2018-01-08
**Acknowledgements revision, December 18**

We would like to acknowledge that we uploaded a revised version of our paper on December 18. Those changes where motivated by the comments of the reviewers and can be summarized in:

1. We added a subsection and a table to compare our method against other popular methods (Adversarial Training and Defensive Distillation).
2. We added a paragraph describing the advantages as well as the disadvantages of using ensemble methods as defense method.
3. We reformulated a few sentenses and fixed a few typos.

We described all this changes in our responses to the reviewers on December 18.

---

### Public Comment · ~Bhavesh_Neekhra1 · 2024-04-30
**Reads well.**

Hi,

Thanks for writing this. I enjoyed reading this paper.

I wonder why was it rejected from ICLR. This paper has 162 citations as of 30 Apr 2024, whereas the mean # of citations for ICLR, 2022 was 48.87 and the median # of citations was 20.

---

### Decision · Program_Chairs · 2018-01-29
**ICLR 2018 Conference Acceptance Decision**

**Decision:**

Reject

**Comment:**

The paper empirically evaluates the effectiveness of ensembles of deep networks against adversarial examples. The paper adds little to the existing literature in this area: an detailed study on "ensemble adversarial training" already exists, and the experimental evaluation in this paper is limited to MNIST and CIFAR (results on those datasets do not necessarily transfer very well to much higher-dimensional datasets such as ImageNet). Moreover, the reviewers identify several shortcomings in the experimental setup of the paper.